# Models of care for Indigenous children in rural and remote settings: A global scoping review

Joseph Freeman[1]*, Thomas Stubbs[1], Verity Chadwick[2], Anita Pickard[1], Tuguy Esgin[3,4,5], Elizabeth J. Elliott[1,6], Alexandra Martiniuk[1,7,8]

1 Faculty of Medicine and Health, University of Sydney, Sydney, New South Wales, Australia, 2 Women and Babies Ambulatory Care, Royal Prince Alfred Hospital, Camperdown, New South Wales, Australia, 3 Faculty of Business and Law, Curtin University, Bentley, Western Australia, Australia, 4 Discipline of Exercise and Sport Science, Faculty of Medicine and Health Sciences, The University of Sydney, Sydney, New South Wales, Australia, 5 School of Medical and Health Sciences, Edith Cowan University, Joondalup, Western Australia, Australia, 6 The Sydney Children's Hospital Network (Westmead), Kids Research, Westmead, New South Wales, Australia, 7 Dalla Lana School of Public Health, University of Toronto, Toronto, Ontario, Canada, 8 George Institute for Global Health, Sydney, New South Wales, Australia

* joseph.freeman@sydney.edu.au

## Abstract

Indigenous communities internationally have demonstrated remarkable strength despite significant challenges. Health disparities among Indigenous peoples persist due to historical injustices and ongoing racial discrimination, not inherent vulnerabilities. Disparities are rooted in a legacy of colonisation, systemic exclusion, and socio-economic inequities impacting access to healthcare, education, and employment. Preliminary searches show limited literature on models of care for remote living Indigenous children. This review aimed to identify internationally, effective models of care for Indigenous children in rural and remote areas. A scoping review was conducted, analysing literature on models of care for remote Indigenous children. This study followed the JBI's Scoping Review Guidance and PRISMA Scoping Review guideline. Inclusion criteria were aged ≤18, rural or remote areas, majority Indigenous, reported health outcomes, described a model of care, in English, and published since 1990. Data were systematically extracted, quality appraised using the 'Aboriginal and Torres Strait Islander Quality Appraisal Tool' then analysed using descriptive-analytical methodology. This review included 16 papers: 8 case series, 3 qualitative studies, and 5 trials. Of these, 7 studies were in Australia, 7 in the USA and 2 in Canada. All studies primary aim was to improve quality of care. Models of care described in the included papers varied, being delivered in traditional healthcare settings, homes, and elsewhere in the community. This review provides insights into the design and implementation of models of care in remote communities with primarily Indigenous populations. The authors recommend 1) that future reviews privilege 'realist evaluation' when examining models of care, 2) designers consider whether a model of care will run for a fixed-period versus ongoing as they have different

**Data availability statement:** All relevant data are within the paper.

**Funding:** This work was supported by the University of Sydney Curtin PhD Scholarship for Clinical Research (to JF); National Health and Medical Research Council of Australia (NHMRC) Investigator Grants to AM (#1195086) and EE (# 2026176); NHMRC partnership funding for Marrura-U (# GNT1171880). The funders had no role in study design, data collection and analysis, decision to publish, or preparation of the manuscript.

**Competing interests:** The authors have declared that no competing interests exist.

requirements for success and 3) a toolkit approach to model of care development like the Qungasvik toolkit which provides evidenced modules adaptable to local conditions, easing workload on local people developing programs.

## 1. Introduction

Many Australian Aboriginal and Torres Strait Islander people (hereafter respectfully referred to as Indigenous people) view health through a holistic lens, encompassing physical, spiritual, social, cultural, and ecological components [1]. This approach emphasises the interconnection between individual health, community well-being, and environmental health. While Indigenous peoples have strengths rooted in a rich culture and history, they continue to face significant challenges stemming from dispossession and ongoing individual and systemic racial discrimination in the wake of colonisation. Despite great strength and resilience, they face barriers to good health. For example, Indigenous Australians have a 2.3 times higher burden of disease than other Australians [2]. They continue to lead innovative, community-driven health initiatives that address these challenges. This is true beyond Indigenous Australia; a large data-linkage study of 28 populations in 23 countries found that Indigenous people globally had poorer health and social outcomes than non-Indigenous people [3]. Although each Indigenous group has distinct language, cultural norms and healthcare needs, they share important similarities; forged from connection to land and experience of colonisation and dispossession [4]. Therefore, sharing global learnings may be useful for improving health outcomes for Indigenous populations.

Indigenous people in rural and remote locations where healthcare access is limited face compounded health difficulties. People living in rural and remote areas also have a higher burden of disease, more limited access to healthcare and shorter life expectancy than urban dwellers. In Australia, the burden of disease is 1.4 times greater in remote areas than in cities [5]. In the USA, people living in rural and remote areas are more likely to be obese and smoke, and more likely to die from heart disease, cancer, unintentional injury, chronic lower respiratory disease, and stroke [6]. These realities highlight the need for culturally informed, community-driven healthcare solutions that leverage Indigenous strengths.

Indigenous people living in rural and remote areas also face insufficient access to culturally appropriate healthcare. A review of health service organisation and access for rural Indigenous people in Canada identified barriers categorised into three groups [7]. The first comprises proximal factors of geography, education attainment and individual racism. Secondly, intermediate factors of employment and income inequalities and inaccessible health and education systems. Lastly, distal factors of colonialism, structural racism, and social exclusion. A similar review of Indigenous child health services in remote Australia showed that services are fragmented both between and within sectors and that innovative models of care are required to provide a unified approach. These models should be guided by local leadership and build upon existing community resources to enhance capacity [8].

The above research indicates that models of care hold the potential to enhance quality, efficiency, and acceptability, even within the context of the increased burden of disease experienced by Indigenous people in rural/remote locations [3]. There is limited literature examining the useful elements of models of care for Indigenous people, especially Indigenous children, which can be applied in the development of novel models of care or to understand effective ways to deliver care which could be expanded or modified in other areas.

This review looks internationally for models of care developed for Indigenous children living in rural and remote areas. The need for this scoping review arose during the development of a health service improvement effort in a remote community in the Kimberley Region of Western Australia [9]. Multiple Indigenous co-authors, including the senior author, helped to centre our research framework on Indigenous principles. However, this review followed non-Indigenous methodological guidance, and this is acknowledged in the limitations section of our manuscript.

Appropriate models of care may improve health systems including their transparency, accountability, evidence-based practice, and clinical governance [10]. Models of care aim to increase quality, improve efficiency, and improve systematic and equitable delivery of health services across populations, ultimately to improve health [10].

Many terms similar to 'models of care' are used in the literature, all of which were included in this study. A 2009 review found 175 overlapping definitions describing models of care, with a range of different objectives [11]. Additionally, a model can be operationalised differently. The following definition of a model of care from an Australian government organisation guided the search criteria development: *'model of care describes the way health services are organised and delivered'* [10]. This definition is expanded to outline how healthcare can be delivered to individuals with coordination of different elements of care as they progress through the stages of a condition, injury, or event [10]. This definition also emphasises that MoC development is a change management process which includes implementation and evaluation [10].

A second literature review of use of the term 'model of care', also published in 2009, provided a more expansive definition of MoC as an '*overarching design for the provision of a particular type of health care service that is shaped by a theoretical basis, evidence-based practice and defined standards. It consists of defined core elements and principles and has a framework that provides the structure for the implementation and subsequent evaluation of care[…]'* [12]. Notably, this definition includes evaluation. The World Health Organisation commonly refers to integrated care models which they acknowledge as loosely defined and roughly synonymous with related terms such as model of care [13]. They consider integrated care in opposition to fragmented and episodic care [13]. These definitions and theoretical frameworks are rooted in Western thinking. Though one might consider some potential overlaps in thinking with Indigenous ways of knowing, being and doing. For instance models of care in Western thinking have similarities with Indigenous ways: being holistic, cyclical, patient/family centred, visually displayed, collaborative, relational and aiming to be fair [14].

'Models of Care' in Indigenous thinking and practice are grounded in a holistic view of health. The National Aboriginal Community Controlled Health Organisation (NACCHO), the peak organisation for Australian Aboriginal controlled health organisations, defines health as *"not just the physical well-being of an individual but refers to the social, emotional and cultural well-being of the whole Community in which each individual is able to achieve their full potential as a human being thereby bringing about the total well-being of their Community. It is a whole of life view and includes the cyclical concept of life-death-life"* [15].

The Australian Indigenous Doctor's Association developed a model of care structured across five dimensions [16]. It is based on existing Aboriginal and Torres Strait Islander definitions of health [17]. These were 1) physical/biological, 2) psychological/emotional, 3) social wellbeing, 4) spiritual, and 5) cultural integrity [16].

A review of Māori-centred models of relational health distilled the models to the following 4 domains: 1) health and wellbeing, 2) whanaungatanga (connectedness), whakawhanaungatanga (building relationships), and socio-political health context (colonisation, urbanisation, racism, and marginalisation) [18]. They noted that health and wellbeing are relational and holistic, and that healthy relations with extended family are imperative [18]. Further, an Australian qualitative study examined the experience of Aboriginal people receiving a diabetes model of care [19]. The participants emphasised to not

limit the model of care to healthcare factors. Instead, they highlighted the importance of incorporating historical components of health and disease (colonisation and family members' experience of diabetes).

In summary, the term 'model of care' is not standardised in the literature and is commonly replaced with many alternate terms.

Our work uses the United Nations definition of Indigenous people: simply, self-identification [20]. This study uses the term 'Indigenous' respectfully to include all First Nations peoples.

This study aimed to determine what models of care exist to support the health of Indigenous children in rural and remote communities. What models have been articulated and how do they work? What benefits do they provide for child health and wellbeing and how they are measured, and what is their community acceptability and cost-effectiveness? The secondary aim was to understand how the concept and term 'model of care' is being used in the literature related to Indigenous children who live in rural/remote communities.

## 2. Methods

### 2.1. Design

This study followed the JBI's updated methodological guidance on scoping reviews [21], and was written using the 20 recommended items of the Preferred Reporting Items for Systematic reviews and Meta-Analyses extension for Scoping Reviews (PRISMA-ScR) Checklist, see S1 Checklist [22]. Though a protocol was developed and iterated, no separate protocol was published or distributed prior to this publication. Currently PROSPERO does not accept scoping reviews. The study design and analysis were influenced by the World Health Organization's health system building blocks [23].

### 2.2. Databases

Databases were searched for peer-reviewed literature in the English language from January 1, 1990 – May 30, 2024. Databases searched were Medline, EMBASE, Web of Science, SCOPUS, and CINAHL.

### 2.3. Search strategy and selection criteria

The search strategy ultimately selected was broad and therefore likely to capture literature with Indigenous worldviews. However, this review was undertaken without a clear published articulation of an Indigenous model of care as guidance. Although the authors sought this, including through meeting with Dr Kate Armstrong the medical advisor to NACCHO, none were found. Principles of Indigenous views of health and health care, as well as understandings of Indigenous ways of knowing, doing, and being were considered throughout the review process.

High-level MeSH headings were used to encompass keywords (e.g., holistic, systems, models, models of care etc.). The MEDLINE strategy is shown in Table 1 and was adapted to all other databases. The complete search strategies are provided in S1 Text.

In this scoping review, studies were included if they met the following inclusion criteria: study participants were 18 years of age or younger (this study uses the term 'child' for those aged 0–18 years), the majority (>50%) participants were Indigenous (self-identified, any country), rural or remote (self-identified, paper used these or synonymous terms to describe study location), publication in the English language, published since 1990, health outcomes stated (though may be delivered in a non-health setting), described a model of care that includes specific patient-level therapy (i.e., not those that focused on measuring disease rather than providing healthcare).

Studies were excluded if they met this criteria: single stage care. That is, the MoC should be multi-faceted and not just contain one component or instance of care delivery, such as single occasion immunisation.

Reference lists of all reviews found in the studies were hand-searched for eligible papers.

This paper uses the term 'model of care'; however, it took an inclusive view of what may be considered a model of care (MoC). It was required only that the model/program be multi-faceted. For example, an outreach Ear Nose Throat (ENT) surgical

**Table 1. MEDLINE search strategy.**

| MEDLINE |
|---|
| ex 'education' OR exp 'ownership' OR exp 'patient reported outcome measures' OR exp 'quality improvement' OR exp 'health services administration OR exp 'quality of life' OR exp 'health care facilities, manpower, and services' OR exp 'patient satisfaction' OR exp 'community mental health services' OR exp 'mental health services' OR exp 'criminology' OR exp 'patient care' OR 'holistic health' OR exp 'evidence-based practice' OR exp 'policy' OR exp 'health planning' OR 'model of care' OR 'care coordination' OR 'wraparound' OR 'first 2'000 days' OR 'outreach' OR 'patient reported' OR 'patient reported experience measures' OR 'delivery of healthcare' OR 'health services' OR 'health care economics' or exp 'health care economics and organizations' OR 'Referral' OR 'collaborative care' OR 'patient centered care' OR 'integrated care' OR 'multidisciplinary care' OR 'intersectoral care' OR 'care pathway' OR 'continuous quality improvement |
| AND |
| Exp 'child' OR exp 'child, preschool' OR exp pediatrics OR exp 'adolescent' |
| AND |
| Exp 'rural health' OR exp 'rural population' OR exp 'rural, hospitals', OR exp 'rural nursing', OR exp 'rural health services' or exp 'remote consultation OR 'rural' OR 'remote' OR 'outback' |
| AND |
| Exp 'Indigenous Canadians' OR exp 'health services, indigenous' OR exp 'Indigenous peoples' OR exp 'oceanic ancestry group' OR exp 'Indians, North America' or 'first nation' OR 'Indigenous' OR 'aboriginal' OR 'australoid' |
| limited to |
| yr = 1990 until May 30, 2024 |
| language = English |

service that provided ENT clinics would not be considered a model as it did not function in relation to the wider health system. However, if the program involved a second element, for example, primary care services and referral to the ENT surgical service or pathways to allied health supports following ENT surgery then it would be considered a MoC for this review. An intersectoral approach, such as integration between justice and health services, was not required for inclusion as a MoC.

## 2.4. Study selection, data extraction, quality appraisal

Title and abstract screening, then full-text review, data extraction and quality assessment were completed within Covidence 2023/24 using predefined data fields (JF,VC). The custom Covidence extraction fields were piloted and iterated. A second reviewer independently validated a portion of the screening and data extraction processes (TS). Resulting discrepancies were resolved through discussion. Studies were quality appraised independently by JF and TS using the Aboriginal and Torres Strait Islander Quality Appraisal Tool [24]. This Australian tool was developed using international research guidelines considering Indigenous epistemologies [24].

Database searching produced 5855 results. These were deduplicated using the Endnote X9 function to 3101 studies. Title and abstract screening decreased the figure to 344 studies and full-text assessment decreased to 16 the number of studies included in this review (Fig 1).

During full-text screening, most excluded studies did not describe a model of care (219 studies). This included studies that provided only a single intervention or phase of work, and studies that focused on disease measurement rather than healthcare provision. Other reasons for exclusion at the full-text screening phase included studies that did not include a majority of Indigenous people (8) and/or were not rural/remote (7) as detailed in Fig 1.

## 2.5. Analytic approach

As a scoping study, this review was iterative and reflexive, and descriptive analyses were focused on predetermined categories/topics for data extraction. The analysis was conducted using a descriptive-analytical method within the narrative tradition, as outlined by Arksey and O'Malley [25]. This approach applies a consistent analytical framework across all included studies, ensuring the systematic collection of standardized information from each study to enhance its usefulness [25]. Consistent with the scoping review data extraction guidelines set by the Joanna Briggs Institute [21], essential details were systematically

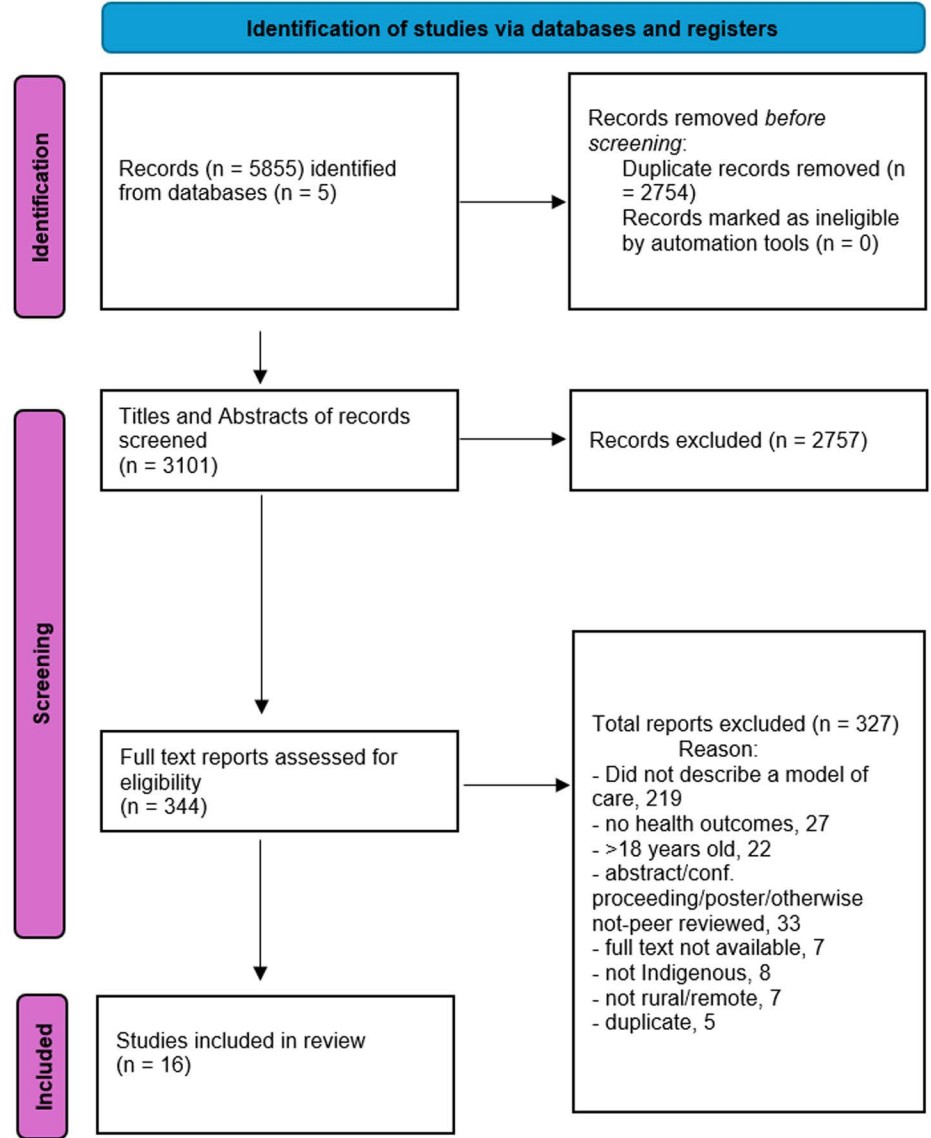

**Fig 1. PRISMA flowchart of study selection for inclusion in the review.**

charted for each article. Data extracted in Covidence software included: title of paper, study aim, problem the model of care aimed to address, location of the study, setting, study design, study population, sample size, age range, and the model of care. Themes were distilled from the extracted data using Microsoft Excel spreadsheets, which were discussed and formed the framework for analysis.

## 3. Results

### 3.1. What models of care were included

This scoping review included 16 papers, summarised in Table 2. There were 8 case series, 3 qualitative studies, and 5 trials. Notably, no studies examined cost-effectiveness. Of these, 7 studies were in Australia, 7 in the USA and 2 in Canada.

All studies stated the primary aim was to improve quality of care, rather than increase cost effectiveness or ease logistical difficulties. Models of care described in the included papers varied, delivered in traditional healthcare settings (including hospitals, outpatient clinics, community health clinics), homes, and elsewhere in the community.

The Aboriginal and Torres Strait Islander Quality Appraisal Tool was used to assess all papers, detailed in Table 3 [24]. A total of 6 of 16 studies responded to a community priority and need while 4 of 16 showed evidence of appropriate community consultation. Only 2 of 16 studies demonstrated Indigenous leadership, and adherence to community protocols. Across all 14 items of the appraisal tool, 47% of responses were 'Unclear'. This demonstrates either inadequate reporting on Indigenous leadership and learnings in these studies or failure to follow best practice for research with Indigenous communities. Only 2 of 16 studies demonstrated the use of an Indigenous research paradigm, only 2 of 16 clearly demonstrated a strengths-based approach, while a further 6 of 16 were 'partially' strengths-based. No studies reported Indigenous ownership of *existing* intellectual property and only 1 reported Indigenous ownership of *created* intellectual property.

Data extraction fields (Table 2) included characteristics of the paper and descriptions of the model of care (MoC). These MoC descriptions are detailed under sub-headings below. This review synthesised examples from the literature into themes.

### 3.2. How do the included models of care work?

The problem addressed was commonly a health condition with high local prevalence including poor oral health [30,37,38,40], diabetes [29,32], Sudden Infant Death Syndrome [41], alcohol and other drug use [27,39], trachoma eye disease [35], mental illness and suicide [27,28,34], and strengthening of the family unit [31]. One study examined health service availability and restructuring [36]. One study described a paediatric outreach service which provides specialist doctors, nurses and occupational therapists to manage acute and chronic disease in the Cape York Peninsula, Australia. Previously, children and caregivers in this region were transferred 800 km to the closest hospital with paediatric services [26].

Participants in the studies had clearly defined entry and exit points within the model of care. All models of care included in this review except the Cape York Paediatric Outreach Service paper [26], captured the entirety of children, in the specific age range, into the model who were in the study area. That is, all children in the specified age range that were physically present in the study area were included. Though some studies targeted specific conditions of high prevalence, all those in the age range were included whether they had the condition of interest or not. Because the studies included groups at risk of an outcome disease as well as those already with the disease, the models of care were both therapeutic and preventative. Only one study was an exception in that it described a general paediatric outreach service which primarily serviced unwell children [26].

In all cases except one [26], the MoC was created specifically for the purpose of the study, and it was not indicated whether the MoC would continue. Agostini et al [26] described a service that existed before study initiation. Therefore, the exit point for participants from the MoC was commonly study completion, or otherwise when they outgrew the included age, or moved from the geographical area.

The workforce cadre in the models of care was reported in three broad categories: Indigenous staff (with varying levels of health training), clinicians, and administrative staff. 'Clinicians' includes Indigenous staff engaged primarily for their clinical skill, while 'Indigenous staff' were Indigenous people engaged primarily for their cultural and community skill. A variety of terms were used for Indigenous staff including Elder and tribal member. Other terms used for Indigenous staff reflected a level of specialisation within health: Indigenous health workers, Aboriginal health workers, Aboriginal oral health aides, local tribal leaders, and family health coaches (Native American community workers). Aside from these roles, the Indigeneity of the general clinicians and other project staff was not reported. This implied mutual exclusivity of the categories of more specialised clinicians (doctors, nurses, dentists, allied health etc.) and Indigenous staff and reflects the lack of Indigenous people, ideally from the relevant cultural and language group, working in highly specialised health roles.

**Table 2. Included studies.**

| First author, year | Title of paper | Study aim | Problem the MoC was trying to address | Location of the study | Setting | Study design | Study population; sample size; age range | The Model of Care |
|---|---|---|---|---|---|---|---|---|
| Agostino 2012 [26] | Cape York Paediatric Outreach Clinic | Describe a MoC (model of care) | Insufficient capacity of local clinics to manage acute medical problems; facilitate management of chronic disease in community | Cape York region, QLD Australia | Local health clinics | Case series | Aboriginal and Torres Strait Islander children in program region; sample and age range not further specified | Outreach paediatric providers to support local clinics in providing acute and chronic treatment, and preventative education. |
| Allen, 2023 [27] | Culturally grounded strategies for suicide and alcohol risk prevention delivered by rural Alaska Native communities: A dynamic wait-listed design evaluation of the Qungasvik intervention | Examine the effectiveness of the Qungasvik (Tools for Life) intervention in enhancing protective factors as a universal suicide and alcohol prevention | High rates of suicide and alcohol use in Alaska Native youth | Rural Yup'ik communities in southwest Alaska | Communities, not otherwise specified | Randomised wait-listed trial | Alaska Native youth; 239; 12–18 years | Utilised 'Qungasvik', a non-prescriptive toolbox through which are designed episodes of Yup'ik cultural engagement. Teachings are at individual, family, or community level and promote 13 factors protective against youth suicide and alcohol use. |
| Allen 2009 [28] | Suicide prevention as a community development process: understanding circumpolar youth suicide prevention through community level outcomes | Describe a methodology for community-level analysis of MoC | Forsaking community-level variables when analysing community-level models | Remote Yup'ik community, Alaska | Community, not further specified | Quasi-experimental | Yup'ik children; 61; 12–17 years | Locally developed and responsive to Elders and season. Utilised 'Qungasvik' (see box above) |
| Brown 2010 [29] | Translating the Diabetes Prevention Program for Northern Plains Indian Youth Through Community-Based Participatory Research Methods | Describe adaptation of an existing MoC | High rates of diabetes risk factors (including overweight) in rural American Indian children | Montana Indian reservations, USA | Clinic, sport facilities, natural environment | Qualitative | Native American and self-identified elder, cultural expert, tribal health worker, educator, parent/guardian, youth, or school food service worker; 31; 10–68 years | Led by elders who identify strategies inherent in their culture, tradition, and environment. Draw heavily on cultural activities already known to the community, including sports, horse-riding, hunting, and gathering. |
| Bryant 2016 [30] | A Community-Based Oral Health Intervention in Navajo Nation Head Start: Participation Factors and Contextual Challenges | Evaluate a MoC | High rates of dental caries | Navajo Nation American Indian reservation in USA | Dental clinic, school | Randomised controlled trial | Parents and children at selected Head Start pre-school programs; 518: mean age 3.6-years, range not given | Oral health promotion program. |

*(Continued)*

**Table 2.** (Continued)

| First author, year | Title of paper | Study aim | Problem the MoC was trying to address | Location of the study | Setting | Study design | Study population; sample size; age range | The Model of Care |
|---|---|---|---|---|---|---|---|---|
| Campbell 2018 [31] | Implementing the Baby One Program: a qualitative evaluation of family-centred child health promotion in remote Australian Aboriginal communities | Evaluate a MoC | Poor general health in remote Indigenous children | Cape York Peninsula, QLD, Australia | Community, not further specified | Qualitative | Apunipima Cape York health workers, family and community members; 48; age range not specified | Family-centred, Indigenous Health worker-led, home-visiting model of care focused on promoting family health to give children the best start to life. It was developed by the Aboriginal community controlled Apunipima Cape York Health Council and delivered in Queensland Cape York remote communities. |
| Chambers 2015 [32] | A Home-Visiting Diabetes Prevention and Management Program for American Indian Youth | Evaluate a MoC | High rates of American Indian youths at risk of, or with type 2 diabetes mellitus | Navajo communities with the White Mountain Apache Tribe, Navajo Nation, USA | Home, clinic, community areas | Cohort | American Indian people, living within a 50-mile radius of health centre, with T2DM, prediabetes, at risk of diabetes (elevated BMI and dyslipidaemia); 255; 0–19 years | Significant home-visiting and family focus (participants were asked to nominate a 'Support Person', usually a family member); developed through 'community-based participatory research'; delivered by local American Indian 'paraprofessionals'. |
| Dimitropoulos 2020 [33] | Outcomes of a co-designed, community-led oral health promotion program for Aboriginal children in rural and remote communities in New South Wales, Australia | Evaluate a MoC | Poor oral health of rural/remote Aboriginal children | 3 communities in Central Northern NSW, Australia | Schools | Case series | Children in the selected schools (all had majority Aboriginal children); 61; 5–12 years | Engagement of local Aboriginal communities in co-design and local delivery by local people. Choose simple models where possible. |
| Etter 2019 [34] | Improving youth mental wellness services in an Indigenous context in Ulukhaktok, Northwest Territories: ACCESS OpenMinds Project | Describe development a MoC | High rates of mental ill-health in Inuit youth | Ulukhaktok, a hamlet of 396 people in the western arctic, Canada | Unspecified | Case series | all Ulukhaktok youth; sample and age range not specified | Human capital development, authentic collaboration, and diversified engagement strategies |
| Ewald 2003 [35] | An evaluation of a SAFE-style trachoma control program in Central Australia | Evaluate a MoC | High rates of trachoma eye disease | Remote community, central Australia (not further specified) | Home, clinic, school | Case series | All Aboriginal children in a remote community; 482; < 13 years | Developed SAFE (Surgery for trichiasis, community Antibiotic programs, Facial cleanliness, and Environmental health improvements), derived from the World Health Organisation approach to trachoma control. |

*(Continued)*

 

**Table 2.** (Continued)

| First author, year | Title of paper | Study aim | Problem the MoC was trying to address | Location of the study | Setting | Study design | Study population; sample size; age range | The Model of Care |
|---|---|---|---|---|---|---|---|---|
| Josif 2017 [36] | The quality of health services provided to remote dwelling aboriginal infants in the top end of northern Australia following health system changes: a qualitative analysis | Evaluate a MoC | Poor health outcomes for remote dwelling Aboriginal infants | Two remote communities in Northern Territory, Australia (not further specified) | Clinic | Qualitative | Aboriginal infants; 25 health workers interviewed; < 1 year | model of care aims to shift focus of primary care from selective to comprehensive. |
| Larsson, 2023 [37] | Improving Early Childhood Caries for American Indian 3- to 5-Year-Old Children Through Interprofessional Outreach: 2018–2022 | Determine impact of dental intervention and health screening on dental health | High rates of untreated decay and long wait time to dental treatment | Northern Cheyenne Nation (rural USA) | School, clinic | Case series | American Indian children; 475; 3–5 years | Component of 'Head Start' child health program consisting of dental education and treatment. |
| Macnab 2008 [38] | 3-Year results of a collaborative school-based oral health program in a remote Indigenous community | Evaluate a MoC | Improved oral health in remote Indigenous children | Hartley Bay, BC, Canada | School | Case series | children in the community from kindergarten to year 10; 13; mean 10 years range not specified | Oral health program designed as 2-way learning for paediatric residents |
| Moore 2018 [39] | Prevention of Underage Drinking on California Indian Reservations Using Individual- and Community-Level Approaches | Evaluate a MoC | High rates of underage drinking by American Indian/Alaska Native youths on rural California Indian reservations | Rural California Indian reservations, USA | Health clinics | Case series | School students from 7th, 9th, and 11th grades; sample size and age range not specified | Motivational interviewing and psychoeducation using a collaborative and non-confrontational approach. Community-level strategy for decreasing underage access to alcohol and reinforcing norms against providing alcohol to youths. |
| Roberts-Thomson 2010 [40] | A comprehensive approach to health promotion for the reduction of dental caries in remote Indigenous Australian children: a cluster randomised controlled trial. | Evaluate a MoC | Poor oral health in remote Indigenous pre-schoolers | Northern Territory of Australia | Community, not further specified | Randomised controlled trial | Indigenous children; 666; 18–47 months | Fluoride varnish was applied to the teeth of each enrolled child. Clinic personnel were trained in dental screening and varnish application and carried out health promotion activities. |
| Young 2019 [41] | Best practice principles for research with Aboriginal and Torres Strait Islander communities in action: Case study of a safe infant sleep strategy | Evaluate a MoC | Sudden Infant Death Syndrome in remote Aboriginal infants | Remote northern regions of Queensland | Home | Case series | Birthing women and their families living in 17 communities in remote northern Queensland; 165; age range not specified | Best practice principles within the Pepi-Pod Program; one community-controlled maternal and child health service that employed an Aboriginal Health Worker-led model of maternal and child health care for remote regions of Queensland. |

**Table 3. Aboriginal and Torres Strait Islander Quality Appraisal Tool [24].**

| Study | Need & Priority | Consultation and Engagement | Leadership | Governance | Protocols | IP & Cultural property protection | IP & cultural property ownership | Data management | Indigenous research paradigm | Strengths-based | Findings to policy & practice | Benefit to communities | Capacity strengthening | Collective learning |
|---|---|---|---|---|---|---|---|---|---|---|---|---|---|---|
| Agostino 2012 [26] | Unclear | Unclear | Unclear | Unclear | Unclear | Unclear | Unclear | Unclear | No | Unclear | No | Unclear | No | No |
| Allen, 2023 [27] | Yes | Unclear | Unclear | Partially | Unclear | No | No | Unclear | Partially | Partially | Unclear | Unclear | Unclear | Unclear |
| Allen 2009 [28] | Yes | Unclear | Unclear | Unclear | Unclear | Unclear | No | Unclear | No | Partially | Unclear | Unclear | No | Unclear |
| Brown 2010 [29] | Partially | Unclear | Unclear | Unclear | Unclear | Unclear | Unclear | Unclear | No | Partially | Partially | Partially | No | Unclear |
| Bryant 2016 [30] | Unclear | Unclear | Unclear | No | Unclear | Unclear | Unclear | Unclear | No | No | Unclear | Unclear | No | Unclear |
| Campbell 2018 [31] | Partially | Unclear | No | Unclear | Unclear | Unclear | Yes | Yes | No | Yes | Partially | Unclear | Unclear | Unclear |
| Chambers 2015 [32] | Unclear | Yes | Unclear | Unclear | Unclear | Unclear | Unclear | No | No | No | No | Partially | Yes | Unclear |
| Dimitropoulos 2020 [33] | Unclear | Yes | Partially | Unclear | Unclear | Unclear | Unclear | Unclear | No | Partially | No | Yes | Partially | Unclear |
| Etter 2019 [34] | Yes | Unclear | Unclear | Unclear | Unclear | Unclear | Unclear | Unclear | Unclear | Partially | Partially | Partially | Unclear | Unclear |
| Ewald 2003 [35] | No | No | Unclear | No | Unclear | No | Unclear | No | No | No | Partially | Unclear | No | No |
| Josif 2017 [36] | No | No | Unclear | No | No | Unclear | Unclear | No | No | No | No | No | No | No |
| Larsson, 2023 [37] | Unclear | Unclear | Unclear | Unclear | No | No | No | No | No | No | Partially | Partially | No | Unclear |
| Macnab 2008 [38] | Yes | Unclear | Unclear | Unclear | Unclear | No | No | No | No | No | Partially | Partially | Partially | Yes |
| Moore 2018 [39] | Unclear | No | No | Unclear | Unclear | Unclear | Unclear | Unclear | No | Partially | Partially | Yes | Unclear | Unclear |
| Roberts-Thomson 2010 [40] | Yes | Yes | Yes | Partially | Yes | Unclear | Unclear | Yes | Yes | No | No | Partially | Yes | Unclear |
| Young 2019 [41] | Yes | Yes | Yes | Yes | Yes | No | Unclear | Yes | Yes | Yes | Partially | Yes | Yes | Partially |

The studies aimed to demonstrate a specific benefit to participants. As per the inclusion criteria, the models of care were required to have at least two components. For example, the multiple components of the model on an oral health MoC were regular application of fluoride varnish, an oral health education program, and supervised toothbrushing. Another study was an Alaskan suicide prevention program, which utilised the 'Qungasvik' toolkit, a modular and multi-faceted package of resources adaptable to different American Indigenous groups and seasons [27,28]. The toolkit was adapted to provide a suicide prevention education session delivered on nature walks and tailored to the season and local customs, as well as to direct alcohol control measures and weekly meetings of the suicide-crisis response group. Another example MoC was The Baby One Program which provided new mothers with a baby basket (including clothing, hygiene items, recipes, etc.), a portable cot designed to reduce SIDS, and in-person health promotion activities [31].

The studies described facilitators of success of the MoC. One study describing an outreach program into Far North Queensland, Australia, benefited from a fly-in-fly-out model with low-turnover of staff [26]. This MoC retained staff in personally and professionally profitable positions in a regional city over a long period, despite remote-working clinicians often being retained for only a few years. The Alaskan suicide prevention program benefitted from the 'Qungasvik' toolkit which provided a basis for flexible MoC delivery. Commonly, workers in Indigenous communities are under-resourced and time-poor with a mixture of professional and social responsibilities [27,28]. A theme from included models of care was being flexible enough to respond to changes in the environment from day to day, adapting to the seasons, the migrating animals, and considering the history of the attending facilitators and participants [27–29,34].

The studies also described challenges. Agostino et al [26] described an outreach service to Far North Queensland with inadequate local staff. They called for increased educational opportunities and career pathways for Indigenous child health workers, proposing that local staff would engage in two-way learning with outreach staff, one side providing western medical expertise, and the other providing continuity of care and cultural knowledge and capital.

However, another study by Etter highlighted the dual identity of community workers and the difficulties they had in providing professional mental health support to friends and family. This is an emotional and cultural burden which comes from working, particularly in mental health, in one's own community, where there can be a high level of familial and cultural connections and obligations which must be balanced with professional responsibilities to provide services and maintain confidentiality [34].

Papers by Allen et al [27,28] proposed use of community-level outcomes in suicide prevention programs highlighting the difficulties in selecting appropriate outcome measures. Currently, a priority for Indigenous health is to use a strengths-based lens. For this reason, Allen et al [27,28] used indirect and strengths-based outcome measures that had previously been associated with a reduced suicide rate. Therefore, they were unable to directly demonstrate change in suicidal ideation or action, but instead measured precursors or factors associated with these.

A diabetes prevention and treatment program in rural USA was adapted to utilise the natural environment, but the researchers reported that they were challenged by severe weather and environmental conditions [29]. The researchers engaged local tribal members with flexible remuneration to lead activities in the outdoors but were challenged with dangerously cold conditions, risk of dangerous wildlife, and lack of transportation options to alternate indoor areas due to snow and ice on the roads.

Two models of care and their evaluations were challenged by a high level of mobility within the relevant Indigenous populations. Ewald et al [35], in their trachoma control program, found that only one third of participants from the target population were resident at all data collection points at 0, 7, and 21 months. The increased mobility of participants led to reduced engagement with the relatively static, place-based program. The program failed to reduce trachoma, and the authors recommended a region-wide approach to capture this mobile population.

Chambers et al [32] conducted a home-visiting diabetes prevention program and encountered a high level of interhousehold mobility. They adapted their program to include a 'support person' who was valuable in locating children for follow-up. Children were asked to nominate their support person at enrolment. They were predominately family members, 70% were

responsible for obtaining medical care for the children and because the children personally selected them, they had influence over the children and their environment. The support person assisted the family as health coaches (American Indian community health workers) to locate and work with the children. The support people were also formally surveyed. The authors concluded that support persons promote and sustain the success of children in a goals-driven health program [32].

Some study authors made recommendations regarding use of their MoC. One study detailed a failed trachoma control program in remote central Australia [35], concluding that their MoC was inappropriate, and recommending a region-wide approach due to high population mobility.

Other studies recommended subsequent adaptation of their MoC initially in neighbouring communities who are likely to be more similar, that increasing numbers of Indigenous and local staff be prioritised, and that two-way learning between Indigenous staff and 'mainstream culture' healthcare workers should be encouraged [26,30–33,38]. These programs also stressed the importance of remaining flexible in future application of the MoC [30], though Bryant et al [30] was an evaluation focused on achieving a high level of fidelity to a pre-existing MoC.

The studies also described knowledge gaps. For example, heterogeneity between even closely neighbouring Indigenous communities can impact upon the utility of adapting models of care. Brown et al [29], who worked in two Montana Indian Reservations in the USA, noted that the Federal government recognised over 550 distinct tribes in the two Reservations, each with their own unique culture, history, and structure. They used community-based participatory research methods to listen to the preferences of those Indigenous people when adapting an existing diabetes prevention MoC. They did not describe how they selected the two of seven Montana reservations and did not note whether they considered the selected communities to be representative of those in Montana, the greater Northern Plains, or more widely.

Etter et al [34] introduced a mental health MoC in Canada. They noted the dual identity required of the community workers when negotiating deep family and professional duties and obligations, and a 'lack of meaningful downtime' [34]. This led to the question of the sustainability of the community workers and therefore the sustainability of the MoC. The impact of this double burden on the feasibility of the MoC was identified as a knowledge gap.

### 3.3. How outcomes were measured

Eleven of 16 studies [26,29,31,33–39,41] were descriptive, following a narrative process of synthesis and analysis. Their methodologies focused on distillation of themes from source material that related to their model of care.

The remaining 5 disease-specific studies used relevant quantitative data, for example standard biometrics of metabolic health for diabetes-focused models, and of oral health for oral-focused models. The outcomes used varied between studies.

Allen et al [27] investigated dose-dependent intervention effects in 2 ultimate and 3 intermediate variables. They were measured at four time points over two years, assessing protective factors against suicide and alcohol risk at individual, family, and community levels. Allen et al [28] used a quasi-experimental design with 2 different survey groups. The first group responded to surveys about community readiness pre- and post-intervention; the second at four time points responding to an evaluation of protective factors, community safety, enforcement, role models, and supports. Results were analysed using a mixed-effects regression model. Bryant et al [29] used bivariable analyses to assess associations between specific child- and parent- factors and number of intervention activities received such as fluoride varnish application. Chambers et al [31] collected participant demographics at baseline, and then intervention process measures included session summaries and satisfaction questionnaires. Outcomes across knowledge, behavior, physiology, and psychosocial domains were assessed at baseline and after 3, 6, and 12 months. Cross-site comparisons were conducted using chi-square tests for categorical variables and one-way ANOVA for normally distributed continuous variables. Roberts-Thomson et al [39] evaluated the impact of the intervention on two secondary endpoints over a two-year period: the extent of community oral health promotion activities and the oral health practices of children. Data were analysed to compute percentages of communities and children reporting 13 health behaviours using Fisher's exact test for health behaviours and t-test for the Oral Hygiene Index and the Gingival Index.

### 3.4. Community acceptability and cost-effectiveness of the MOC

Studies were rated for community acceptability using the Aboriginal and Torres Strait Islander Quality Appraisal Tool [24]. A minority of papers demonstrated Indigenous leadership or consultation. Only 4 of 16 papers demonstrated evidence of community acceptability. One included study of oral health in Australia anonymously surveyed parents and found evidence for acceptability of their oral health MoC [33].

## 4. Discussion

This review included 16 global publications in children, restricted to the English language, which enrolled Indigenous participants from Australia, Canada, and the USA. This review shares learnings relevant to all countries with a settler-colonial history and ongoing discrimination of Indigenous people. Australia, Canada, and the USA are settler colonies, with comparable socio-political histories. The governments of these countries retain a large part of responsibility centrally for resources and policy direction for Indigenous healthcare. Like Australia [42], Canada and the USA have shifted from assimilation to self-determination since the 1970s. Unlike Australia, however, both Canada and the USA have formal treaties with their Indigenous populations [43].

Our review did not require a specific term such as 'model of care' be used in papers for inclusion; however, it did require that a MoC be described. A precursor MEDLINE search of *model of care[title] AND pediatric[title/abstract] OR child[title/abstract]* returned only 50 results. This demonstrated that the specific term 'model of care' is seldom used. Therefore, this study used a far wider search strategy and looked for elements that constitute a MoC. This study was restrictive in other ways, in that it focused on Indigenous children in rural or remote settings. The limited literature found demonstrates a lack of existing peer-review literature providing description, analysis and discussion of MoC globally.

Health service delivery in remote areas is challenged by 'diseconomies of scale' with small, dispersed populations [44]. This highlights the need for efficient integration of health services [44]. However, most included studies did not attempt to place their program within a wider context of the health system, although the study by Agostini et al [26] partially described the interface with the wider health system. They provided a brief history of the Cape York Paediatric Outreach Clinic and described how their service provided early follow-up of children discharged from large regional hospitals, as well as seeing regular and acute community patients. However, Agostini et al did not describe the specific process of referral or interface, for example how patients are handed over and how they escalate care.

The limited number of included studies, and the fact that none were wide-ranging, ongoing, or illustrated within the context of a wider health system, shows a gap in the academic literature as well as likely in practice. Existing literature suggests that with increasing remoteness, health services should become increasingly integrated to aggregate a critical service population mass [44,45]. For example, none of the included studies provided a thorough account of how clients in a MoC could be referred in and out from a wider health system. Rather, all but one study [26] included all children present in the study location at risk of a condition without contextualising their health care environment. Also, while research suggests that multifactorial risk factors for a single disease also increase risk for multiple diseases [46], no study discussed the care of the 'study condition' in the context of other conditions.

With so many concurrent components, intervention studies aiming to isolate one effect are difficult or impossible. Existing research suggests that increased complexity requires increased detail in reporting outcomes [47]. Even descriptive studies of models of care defy usual methods when there are many permutations of any one individual client's experience through the care components on offer within a model. Methodology for systems-level analysis is less defined and contains high levels of uncertainty compared to analysing small, definable parts of a health system [47]. This 'reductionist' approach reflects current analytical limits of complexity and uncertainty, hurdles which may be overcome with new methodologies utilising big data and AI-informed approaches.

When adapting existing models of care to new settings, fidelity to the existing MoC versus flexibility and freedom of adaptation to fit the target community was a common theme in included papers. Some included studies called for

subsequent adaptation of their own MoC [30,33,38]. Existing research shows that it can be unclear if the effective components of complex interventions are successfully transferred to a new program, or if those components are effective in a new context [48]. In contrast, flexibility in MoC development is critical as Indigenous knowledge systems and lifestyles are highly linked to environment [41]. Bryant et al [30] note that flexible interventions are more able to overcome challenges.

Other authors raised the more fundamental question of the appropriateness of adaptation, or whether models of care should be developed from the ground up in each place at each time [28,49]. There are almost always important differences between settings, whether that be in terms of culture, language, logistics, organisational behaviour, staff roles, community and family expectations, history and so on. In terms of culture, Okamato et al [50] describe a 'culturally grounded approach' which favours 'ground up' development which they define as arising 'from the values, beliefs, and worldviews of the children that are the intended consumers of the program'. The broad concept is well discussed in literature and referred to as the 'fidelity-adaptation dilemma' [51,52].

Conversely, if the essential elements and processes leading to the desired change are understood, adapting them to local context might not change the efficacy of a MoC. There is difficulty in determining whether the adapted components maintain their original effectiveness without additional evaluation [48]. Also, delivering an intervention exactly as it was originally designed in a new context will inherently yield different results—and may alter its efficacy, or have adverse effects.

A toolkit approach can provide a method of adapting evidence-based modules to another context in a time efficient and context sensitive manner. Allen et al [28] described their use of the Qungasvik toolkit which provides evidence-based mental health modules and program components which can be adapted to local conditions, thus easing the workload on local people to develop or adapt a MoC. It is strengths-based and designed to give people aged 12–18 years 'reasons for sobriety' and 'reasons for life' [53]. The Qungasvik approach balances fidelity and flexibility. Toolkits to ease the delivery of evidence-based models of care exist, including in substance use [54], cancer care [55], and diabetes care [56].

This review identified an important distinction between models of care intended to run for a fixed period compared to models of care designed to be ongoing. For example, Ewald et al [35] conducted a study that was designed to cease after the transmission of trachoma reduced, while Roberts-Thomas et al [40] intended their oral health campaign to be ongoing, pending positive evaluation. Both 'fixed period' and 'ongoing' models of care require co-leadership and design and community goodwill, because health work done by Indigenous people will likely be more effective as they are likely to have greater access to community knowledge and communication skills and hence greater cultural knowledge [3]. Ongoing models of care rely on ongoing good-will, two-way learning, and continuous quality improvement, and there is a greater incentive to transfer skills and governance to community members to enhance efficacy and sustainability. Learnings from an'ongoing' MoC will be more applicable to a future 'ongoing' MoC.

Models of care may simply be understood as how health services are delivered. However, this study did not identify a consistent or clear articulation of what a 'model of care' definitively entails. Instead, the research highlighted the presence of various types of models, suggesting that models should be designed with flexibility to accommodate diverse settings, needs, and preferences. There is a need for Indigenous led, culturally secure healthcare models for Indigenous children in remote areas. These should be co-designed with Indigenous people, based on community priorities, and evaluated using Indigenous research paradigms. Indigenous input to address systemic challenges inherent in remote services will result in models that benefit Indigenous children are sustainable and shift the discourse from deficit-based models to strength-based Indigenous solutions.

Although this paper acknowledges the roles of colonization and systemic racism in shaping Indigenous child health outcomes, a deeper examination of how colonial healthcare structures persist is essential. Historically, colonial policies such as residential schools, forced relocations, and discriminatory healthcare practices have disrupted Indigenous health systems and eroded trust in medical institutions. Today, these legacies continue through systemic barriers, including limited access to culturally safe care, healthcare provider bias, and the underfunding of Indigenous health services [57].

This review had limitations. Firstly, this paper and its thinking comes largely from a settler-colonial country, Australia, and its institutions. Therefore, the forces and traditions of colonisation and structural racism which continue to profoundly negatively shape Indigenous child health outcomes, also confine its analysis here. Notably however, multiple authors of this paper are Indigenous with attendant non-settler-colonial lived history and experience. The limited scope of the literature challenged synthesis. Additionally, the captured literature often described a MoC only as a secondary aim, thus limiting the provided description and analysis of the model itself. The English only search strategy may have limited results given a lack of included European or Asian studies. Additionally, the variation in use of terms to describe models of care possibly reduced included studies, although this was countered with a very wide search strategy.

This review provides insights into the design and application of models of care in remote communities where the population is mostly Indigenous and informs our recommendations. The authors recommend that future reviews privilege 'realist evaluation' when examining models of care for Indigenous children, as more become available. Realist evaluation is an alternate (or additional) methodology that may be usefully applied in evaluations of models of care. This method examines the basic evaluative question 'does it work', in a more nuanced way by asking 'what works for whom, in what contexts, in what respects, and how?' [58]. None of the included studies used realist evaluation methodology. However, beyond what emerged from the existing and limited literature, we recommend use of Indigenous evaluation frameworks such as the Indigenous Evaluation Framework [59], or the Indigenous Advancement Strategy Evaluation Framework [60].

## 5. Conclusion

This scoping review identified models of care, and their evaluations, for Indigenous children in remote communities. The findings emphasise the importance of differentiating between fixed-term and ongoing models of care due to their varying success factors. A toolkit approach, like the Qungasvik toolkit [28] is recommended to facilitate adaptable and locally relevant model development. Toolkits are a recognised method of balancing fidelity vs adaptation [61] and facilitating co-design and local Indigenous-leadership of models of care while efficiently using existing 'building-blocks' and balancing limited resources [62].

## Supporting information

**S1 Text. Appendix 1.**
(DOCX)

**S1 Checklist. PRISMA checklist.**
(DOCX)

## Acknowledgments

We pay our respects to the Traditional Custodians of Noongar Whadjuk Boodjar - the longest living and most resilient culture globally, of which this paper's Aboriginal author is part. This is the land in which the largest Aboriginal community in the Australia still resides and thrives. We respectfully recognise Elders past, present and emerging - their flora and fauna, spiritual beliefs, songlines and Dreamtime stories of creation. We respectfully recognise game and food sources, water sources, shelter, medicine and bush tucker.

We also offer our acknowledgements to the Indigenous Peoples of all countries represented in the paper including, but not limited to, First Nations, Inuit and Métis Nations in Canada, Native American, and Alaska Native and Native Hawaiian nations in the United States. We pay our respects to their cultures, their connection to the land, and the resilience they bring to their community.

We also pay our respects to the spirit of the Indigenous and non-Indigenous authors upon whose work this review was built, to the spirit of all those whose lives and work we honour through our commitment to advocate for the rights,

wellbeing and health of Indigenous peoples around the world, and to the spirit of all of those who lay the foundation for culturally responsive and equitable systems of care.

## Author contributions

**Conceptualization:** Joseph Freeman, Elizabeth J. Elliott, Alexandra Martiniuk.

**Data curation:** Joseph Freeman, Thomas Stubbs, Alexandra Martiniuk.

**Formal analysis:** Joseph Freeman, Alexandra Martiniuk.

**Funding acquisition:** Joseph Freeman, Elizabeth J. Elliott, Alexandra Martiniuk.

**Investigation:** Joseph Freeman, Verity Chadwick, Anita Pickard, Elizabeth J. Elliott, Alexandra Martiniuk.

**Methodology:** Joseph Freeman, Thomas Stubbs, Tuguy Esgin, Elizabeth J. Elliott, Alexandra Martiniuk.

**Project administration:** Alexandra Martiniuk.

**Supervision:** Tuguy Esgin, Elizabeth J. Elliott, Alexandra Martiniuk.

**Validation:** Joseph Freeman, Thomas Stubbs.

**Writing – original draft:** Joseph Freeman.

**Writing – review & editing:** Joseph Freeman, Thomas Stubbs, Verity Chadwick, Anita Pickard, Tuguy Esgin, Elizabeth J. Elliott, Alexandra Martiniuk.

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
