## [Decision Letter · Decision Letter 0]

PGPH-D-24-02491

Models of care for Indigenous children in rural and remote settings: a global scoping review

Dear Dr. Freeman,

Thank you for submitting your manuscript to PLOS Global Public Health. After careful consideration, we feel that it has merit but does not fully meet PLOS Global Public Health’s publication criteria as it currently stands. Therefore, we invite you to submit a revised version of the manuscript that addresses the points raised during the review process.

Academic Editor Guillaume Fontaine, PhD: Thank you for your submission. In addition to the reviewers' comments, please address the following points in your revision:

1. Introduction: Please remove the sub-headings in this section, these are not necessary. Please review to insure that each paragraph of the introduction is well defined, and flows well to the next one. Sentences like "the paper chooses the term..." are usually not seen in the introduction and belong to the methods when defining concepts.

2. The PRISMA-ScR is a reporting checklist, and not methodological standards. Were you also guided by scoping review methodological standards (e.g., JBI, Arksey and O'Malley).

3. Please attached all search strategies used as appendices.

4. The analysis section is very short, and it is unclear exactly what was done. Please expand this section and give concrete examples.

5. The section "How outcomes were measured" is insufficiently detailed. It would be relevant to include descriptive statistics on the frequency of different outcome measures, more extensive descriptions of these measures, and examples.

We look forward to receiving your revised manuscript.

Kind regards,

Guillaume Fontaine, PhD, RN

Academic Editor

Journal Requirements:

Additional Editor Comments (if provided):

N/A

Reviewers' comments:

Reviewer's Responses to Questions

**Comments to the Author**

1. Does this manuscript meet PLOS Global Public Health’s publication criteria?

Reviewer #1: Partly

Reviewer #2: Yes

Reviewer #3: Yes

Reviewer #4: Partly

2. Has the statistical analysis been performed appropriately and rigorously?

Reviewer #1: N/A

Reviewer #2: Yes

Reviewer #3: Yes

Reviewer #4: N/A

3. Have the authors made all data underlying the findings in their manuscript fully available (please refer to the Data Availability Statement at the start of the manuscript PDF file)?

Reviewer #1: Yes

Reviewer #2: Yes

Reviewer #3: Yes

Reviewer #4: Yes

4. Is the manuscript presented in an intelligible fashion and written in standard English?

Reviewer #1: Yes

Reviewer #2: Yes

Reviewer #3: Yes

Reviewer #4: Yes

Reviewer #1: The article addresses a critical and under-researched topic. While it has merit, the methodological section requires improvement. The identified methodological limitations raise concerns about potential biases and the external validity of the findings. Furthermore, the argumentation could benefit from better sequencing of cause-effect links and more robust referencing to support key claims.

Reviewer #2: *the paper acknowledges Indigenous perspectives on health but it does not center them in the study's framework. It would be beneficial to explicitly incorporate an Indigenous-led research paradigm to reinforce self-determination.

*conclusion could be strengthened by advocating for co-designed, Indigenous-led models of care rather than externally imposed ones

*the paper acknowledges colonisation and systemic racism, however the discussion would benefit from a stronger critique of how colonial healthcare structures continue to shape Indigenous child health outcomes

*The paper recommends realist evaluation for future reviews. While useful, this is still a Western evaluation framework, next time consider advocating for Indigenous evaluation frameworks.

This is an important and well-structured paper that amplifies the need for Indigenous-led, culturally secure healthcare models for Indigenous children in remote areas. By strengthening its methodological grounding in Indigenous research paradigms, reinforcing Indigenous leadership, and deepening the discussion of systemic challenges, this paper has the potential to shift the discourse from deficit-based models to strength-based Indigenous solutions.

Reviewer #3: Well done to the authors.

While you spent some time talking about how different entities frame and operationalize model of care, your manuscript would benefit from an articulation of what model of care entails exactly, at least from your own perspective after consulting all of these sources.

Also, you might want to pay attention to your formatting and keep it consistent. In some cases, you start with 'six' of 16 studies... and then continue with ‘4’ of 16 studies in the same sentence. You should choose to either spell out the numbers and keep that consistent or use their digit.

Reviewer #4: Title: Models of care for Indigenous children in rural and remote settings: a global scoping review

Finding and assessing successful care models for Indigenous children in rural and isolated places is the goal of this assessment. This research aims to inform the creation of contextually relevant and culturally suitable healthcare models that cater to the special requirements of Indigenous children by drawing on experiences from throughout the world.

The use of the pronoun (we) is not encouraged, so please rephrase the sentence.

participants were less than or equal to 18 years of age (child is used to refer to this age group): Needs more clarifications as childhood (3 to 11 years old), adolescence, or teenage (from 12 to 18 years old)?

In the text, when referring to the author et al., please include the No. of references following the author et al.

Where is the statistical analysis?

Please include a few lines explaining the conclusion of this study.

Grammar: Needs minor corrections throughout the manuscript.

**Do you want your identity to be public for this peer review?** For information about this choice, including consent withdrawal, please see our Privacy Policy

Reviewer #1: No

Reviewer #2: No

Reviewer #3: No

Reviewer #4: **Yes: ** Sura Saad Abdullah

---

## [Editor Report · Decision Letter 1]

PGPH-D-24-02491R1

Models of care for Indigenous children in rural and remote settings: a global scoping review

Dear Dr. Freeman,

Thank you for submitting your manuscript to PLOS Global Public Health. After careful consideration, we feel that it has merit but does not fully meet PLOS Global Public Health’s publication criteria as it currently stands. Therefore, we invite you to submit a revised version of the manuscript that addresses the points raised during the review process.

Thank you for submitting the revised version of your manuscript. Please address the following points:

1. The methods section of the abstract is underdeveloped; consider cutting back the background and conclusion of the abstract to add to the methods what methodological guidelines were followed, eligibility criteria, study selection and data extraction process and data analysis methods.

2. Sources: You conducted searches in different bibliographical databases. Were other sources searched, considering this is a scoping review? For example, grey literature sources (Google Scholar, Open Grey), government reports, dissertations and theses, hand-searching of relevant journals? If these were not included as sources, why?

3. Sources: The sentence "High-level MeSH headings were used to encompass keywords (e.g., holistic, systems, models, models of care etc.). The MEDLINE strategy is shown in Box 1 and was adapted to all other databases. The complete search strategies are provided in Appendix 1." goes into search strategy.

We look forward to receiving your revised manuscript.

Kind regards,

Guillaume Fontaine, PhD, RN

Academic Editor

Journal Requirements:

Additional Editor Comments (if provided):

N/A
---

## [Editor Report · Decision Letter 2]

Models of care for Indigenous children in rural and remote settings: a global scoping review

PGPH-D-24-02491R2

Dear Dr Freeman,

We are pleased to inform you that your manuscript 'Models of care for Indigenous children in rural and remote settings: a global scoping review' has been provisionally accepted for publication in PLOS Global Public Health.

Best regards,

Guillaume Fontaine, PhD, RN

Academic Editor